# Intersection of gambling with smoking and alcohol use in Great Britain: a cross-sectional survey in October 2022

Loren Kock [ID] ,[1,2] Sharon Cox,[1,2] Lion Shahab,[1,2] Amanda Roberts,[3] Steve Sharman,[4] Vera Buss,[1,2] Jamie Brown[1,2]

¹Department of Behavioral Science and Health, University College London, London, UK
²SPECTRUM Research Consortium, Edinburgh, UK
³School of Psychology, University of Lincoln, Lincoln, UK
⁴National Addiction Centre, King's College London, London, UK

**Correspondence to**
Dr Loren Kock; l.kock@ucl.ac.uk

## ABSTRACT

**Objectives** Gambling is associated with cigarette smoking and alcohol consumption. We explored the intersection of gambling across all risk levels of harm with smoking and alcohol use among adults in Great Britain.

**Design** A nationally representative cross-sectional survey in October 2022.

**Setting** Great Britain.

**Participants** A weighted total of 2398 adults (18+ years).

**Outcome measures** We examined the prevalence of past-year gambling and, among those reporting gambling, assessed the associations between the outcome of any risk of harm from gambling (scoring >0 on the Problem Gambling Severity Index) and the binary predictor variables of current cigarette smoking and higher risk alcohol consumption (AUDIT-C score≥4). We also explored data on weekly expenditure on gambling with smoking and alcohol use among those categorised at any-risk of harm from gambling.

**Results** Overall, 43.6% (95% CI 41.2% to 45.9%) of adults gambled in the past year. Among these, 7.3% (95% CI 5.3% to 9.3%) were classified at any-risk of harm from gambling, 16.0% (95% CI 13.2% to 18.8%) were currently smoking and 40.8% (95% CI 37.2% to 44.4%) were drinking at increasing and higher risk levels. There were no associations between any risk of harm from gambling and current smoking (OR adjusted=0.80, 95% CI 0.35 to 1.66) or drinking at increasing and higher risk levels (OR adjusted=0.94, 95% CI 0.52 to 1.69), respectively. Analyses using Bayes factors indicated that these data were insensitive to distinguish no effect from a range of associations (OR=95% CI 0.5 to 1.9). The mean weekly spend on gambling was £7.69 (95% CI £5.17 to £10.21) overall, £4.80 (95% CI £4.18 to £5.43) among those classified as at no risk and £45.68 (95% CI £12.07 to £79.29) among those at any risk of harm from gambling.

**Conclusions** Pilot data in a population-level survey on smoking and alcohol use yielded similar estimates to other population-level surveys on gambling participation and at-risk gambling. Further data are needed to elucidate the intersections more reliably between gambling, smoking and alcohol use and inform population-level approaches to reduce harm.

## STRENGTHS AND LIMITATIONS OF THIS STUDY

⇒ The main strength of this study is the inclusion of measures of gambling behaviour in a long-running representative survey of the smoking and alcohol use in Great Britain.

⇒ Limitations are that the data are cross-sectional and self-reported, and a small number of respondents were classified as at any-risk of harm from gambling.

⇒ There is uncertainty in our estimates for expenditure on gambling among those at any risk of harm from gambling, but the upper limits remain plausible for disordered gambling behaviour.

in the previous 12 months.[1 2] Although many individuals gamble as a social activity without experiencing harm, some gamble at levels that adversely impact financial stability, personal and family well-being and physical and mental health.[3 4] Conservative estimates indicate that approximately 0.3–0.5% of the general UK adult population report severe gambling behaviours that warrant a diagnosis of gambling disorder (hereafter termed 'disordered gambling') and 3–4% are 'at-risk' (those who experience a low or moderate level of problems leading to some negative consequences, and relative to disordered gamblers drive most of the harm from gambling at the population level).[2 5–7] Due to a combination of financial and health costs associated with gambling (including homelessness, suicide, depression, alcohol dependence, illicit substance use, unemployment and imprisonment), gambling is also associated with an estimated annual economic cost to the UK government of ~£413 million and potentially £655 to £1355 million in societal value due to adverse health effects.[8]

Gambling is thought to be associated with other addictive behaviours, including cigarette smoking and excess substance use such as alcohol consumption,[9–11] and this may relate to common neurobiological,

## INTRODUCTION

General population surveys estimate that half of adults in Great Britain have gambled

genetic and socio-environmental factors which could act to reinforce each respective behaviour.[12–14] Previous prevalence surveys in Great Britain have illustrated mixed evidence on the relationship between disordered gambling and the use of these substances. The 2007 British Gambling survey indicated that smoking was associated with higher rates of past-year gambling (79% vs 64% in those who did not smoke) and disordered gambling (1.4% vs[15] 0.4%), while the prevalence of disordered gambling was 3.4% in those who consumed >20 units of alcohol on their heaviest drinking day and 0.1% who consumed 1–4 units.[16] In contrast, a 2021 evidence review on gambling-related harms conducted by Public Health England concluded that although increased alcohol consumption was associated with gambling at all levels of harm, there was no apparent association with smoking.[2]

The co-occurrence of gambling with smoking and increasing and higher risk alcohol consumption is important to study at the population level in the context of public health and health inequalities. It is likely that co-occurrence of these behaviours compound the physical, social, financial and psychological harms that each of them causes. These harms may be disproportionately greater for certain subgroups, namely those experiencing poverty[17] who are also more likely to smoke and experience greater harms from alcohol consumption compared with more advantaged groups.[18 19] Due to their high relative costs, expenditure on smoking and alcohol can exacerbate and push low-income households into poverty.[15] Likewise, money spent on gambling as a proportion of total expenditure may be higher in less advantaged households.[17] Expenditure on all three products is of concern particularly in the UK and elsewhere where rising prices for everyday items and services have resulted in less 'disposable' income, particularly among lower-income households.[20] Since these individuals are less able to absorb the added burden of this 'cost of living' crisis, it follows that less advantaged groups suffer greater psychosocial and material harm than more advantaged households even in the absence of the harms caused by gambling and substance use behaviours.[21]

Smoking, alcohol and gambling among adults aged 18+ are currently legal in the UK, with the highly profitable underlying industries regulated to different degrees by the UK government but with similar motives to disrupt policies seeking to reduce the harm from the use of their products.[22] Data from a representative sample of adults can provide insight into the dynamics of these behaviours—for instance the potential to substitute or complement one with another[23 24]—in an evolving sociocultural and regulatory context. To build on the existing research reporting on these three behaviours, we piloted the addition of several standard gambling measures to an ongoing representative monthly survey of smoking and alcohol use in Great Britain. The objective of this study is to explore the intersection of gambling across all risk levels (henceforth termed 'any-risk' gambling) with smoking and alcohol use among adults in Great Britain. Specifically, we aimed to (1) estimate the prevalence of past-year gambling according to smoking and increasing and higher risk alcohol use, (2) assess the associations between any-risk gambling (defined by scores of >0 on the Problem Gambling Severity Index) and smoking and increasing and higher risk alcohol use, respectively and (3) explore the average weekly expenditure on gambling, smoking and alcohol use among those reporting any-risk gambling.

## METHODS
### Sample and recruitment
The study population consisted of adults aged ≥18 and over living in households in Great Britain, surveyed in October 2022 in the Smoking and Alcohol Toolkit Study (STS/ATS). Ethical approval for the STS/ATS was granted by the University College London Research Ethics Committee (ID 0498/001). The data were not collected by UCL and were anonymised when received by the research team. In accordance with ethical approval, all respondents provided informed verbal consent.

The STS/ATS uses a hybrid of random location and quota sampling to select a new sample of approximately 2400 adults (aged ≥16 years) each month in Great Britain. Telephone interviews are carried out with one household member until quotas based on factors influencing the probability of being at home (eg, gender, age, working status) are fulfilled. We used survey weighting to match descriptive data to socio-demographic profile in Great Britain (based on age, social grade, region, tenure, ethnicity and working status within sex). Detailed survey methodology is reported elsewhere.[25 26] Comparisons with sales data and other national surveys show that the STS recruits a representative sample of the population in Great Britain with regard to key demographic variables and smoking indicators.

For the current study, all adults were asked a question pertaining to past year gambling participation (derived from indications of type of gambling). Due to funding constraints, questions used to derive the Problem Gambling Severity Index and weekly expenditure on gambling were asked to a partial sample consisting of ~88% of the total eligible sample of people who indicated that they gambled in the past year.

### Measures
The measures included in the current study are summarised in table 1 below. Full details on items used to code these variables are provided in the online supplemental appendix.

### Patient and public involvement
None.

**Table 1** Summary of measures

| Measure | Description | Variable type |
|---|---|---|
| Past-year gambling (categorical) Levels: yes; no) | Affirmative responses to any of the below gambling types in the past year: ▶ National lottery, other lotteries or scratch cards. ▶ Football pools. ▶ Bingo (not online). ▶ Slot machines. ▶ Machines in a bookmakers. ▶ Casino table games (not online). ▶ Online gambling in slots, casino or bingo. ▶ Online betting with a bookmaker. ▶ Betting exchange. ▶ Horse races (not online). ▶ Dog races (not online). ▶ Sports events (not online). ▶ Private betting. ▶ Loot boxes or skins gambling within online/video games. ▶ Crypto casinos. ▶ Any other gambling event or activity. ▶ Have not done any of these things. | Binary stratification variable |
| Problem Gambling Severity Index (categorical) Levels: no risk; low risk; moderate risk; disordered gambling | The PGSI is a nine-item questionnaire on gambling severity and was derived from the Canadian Problem Gambling Index,[39 40] and asked to those categorised as having gambled in the past year (see online supplemental appendix for full item list). Scores between 1 and 7 represent 'at risk' gambling (1–4 'low risk' and 5–7 'moderate risk'). An individual scoring 8 or higher is classified as a 'disordered gambler'.[41] | Outcome |
| Any risk of harm from gambling (categorical) Levels: no risk; any risk | A category of 'any-risk' refers to those scoring 1 or greater on the PGSI. | Outcome (recode) |
| Weekly expenditure on gambling, smoking and/or alcohol (continuous: in GBP (£)) | Weekly expenditure on gambling, smoking and alcohol was derived from responses to three questions regarding expenditure on each. | Outcome |
| Smoking status (categorical) Levels: currently smoking; not smoking | Respondents were classified according to whether they were currently smoking cigarettes (smoke every day; smoke but not every day) or not (do not smoke cigarettes but smoke tobacco of some kind; stopped smoking in the last year; stopped smoking more than 1-year ago; never smoked). | Predictor |
| Level of alcohol consumption (categorical) Levels: <4 on AUDIT-C; ≥4 on AUDIT-C (drinking at increasing and higher risk levels) | Level of alcohol consumption in the last 6 months was assessed using the consumption items from the Alcohol Use Disorders Identification Test-Concise (AUDIT-C),[24] a three-item screening tool developed by the WHO, with a score ranging from 0 to 12. Respondents scoring 4 or higher on the AUDIT-C were classified as drinking alcohol at increasing and high-risk levels. | Predictor |
| Social grade (categorical) Levels: ABC1; C2DE | Social grade based on occupation (ABC1: higher and intermediate managerial, administrative and professional, supervisory, clerical and junior managerial, administrative and professional; C2DE: skilled manual workers, semi-skilled and unskilled manual workers and state pensioners, casual and lowest-grade workers, unemployed with state benefits).[42] | Covariate |
| Age (categorical) Levels: 18–24; 25–34; 35–44; 45–54; 65+ | Age in years at the time of the survey. | Covariate |
| Sex (categorical) Levels: women; men; in another way/refused | Identified sex at the time of the survey. All response options were reported in sample characteristics, but due to small case numbers of 'in another way/refused' this category was excluded from regression analyses), and. | Covariate |
| Region in GB (categorical) Levels: North, Midlands, South Scotland; Wales | Region in England at the time of the survey. | Covariate |

GB, Great Britain; PGSI, Problem Gambling Severity Index.

## Analyses

The analyses were pre-registered on the Open Science Framework, https://osf.io/nc6jm and conducted in R V.4.2.2 (packages *tidyverse* and *survey*[27 28]) with all statistical code made open-access at https://osf.io/aj7c9/. The study followed the Strengthening the Reporting of

**Table 2** Characteristics of the sample

| Characteristic | Overall n=2398 | Did not gamble n=1353 | Gambled in past year n=1045 |
|---|---|---|---|
| **Age** | | | |
| 18–24 | 13.7%, (328) | 16.2%, (219) | 10.5%, (110) |
| 25–34 | 16.7%, (400) | 16.3%, (221) | 17.2%, (180) |
| 35–44 | 15.4%, (370) | 15.0%, (203) | 16.0%, (168) |
| 45–54 | 16.5%, (395) | 14.8%, (201) | 18.6%, (194) |
| 55–64 | 15.0%, (359) | 12.1%, (164) | 18.6%, (194) |
| 65+ | 22.7%, (544) | 25.5%, (345) | 19.1%, (199) |
| Missing | 2 | 2 | 0 |
| **Sex** | | | |
| Men | 48.7%, (1157) | 46.5%, (622) | 51.5%, (535) |
| Women | 50.8%, (1209) | 53.1%, (710) | 47.9%, (499) |
| In another way | 0.5%, (12) | 0.4%, (6) | 0.6%, (6) |
| Missing | 20 | 15 | 5 |
| **Social grade** | | | |
| AB | 26.1%, (626) | 26.4%, (357) | 25.8%, (269) |
| C1 | 29.9%, (716) | 29.5%, (399) | 30.3%, (316) |
| C2 | 20.3%, (486) | 19.9%, (270) | 20.7%, (216) |
| D | 14.5%, (348) | 14.2%, (193) | 14.8%, (156) |
| E | 9.3%, (222) | 9.9%, (134) | 8.4%, (88) |
| **Region** | | | |
| South | 36.6%, (878) | 36.1%, (489.2) | 37.2%, (388.3) |
| Midlands | 25.9%, (622) | 25.5%, (345.7) | 26.5%, (276.5) |
| North | 23.8%, (571) | 20.6%, (279.3) | 27.9%, (291.6) |
| Wales | 4.9%, (118) | 6.2%, (84.0) | 3.3%, (34.1) |
| Scotland | 8.7%, (209) | 11.5%, (155.2) | 5.2%, (54.1) |
| **PGSI category** | | | |
| Did not gamble | 56.4%, (1353) | 100.0%, (1353) | – |
| No risk | 40.4%, (969) | – | 92.7%, (968.5) |
| Low risk | 2.6%, (62) | – | 5.9%, (62) |
| Moderate risk | 0.5%, (11) | – | 1.0%, (11) |
| Disordered gambling | 0.1%, (3) | – | 0.3%, (3) |
| **Smoked cigarettes** | 14.5%, (345) | 13.4%, (181) | 16.0%, (164) |
| Missing | 26 | 8 | 18 |
| **AUDIT-C 4 or higher** | | | |
| 4 or higher | 33.4%, (775) | 27.7%, (359) | 40.8%, (415) |
| Missing | 82 | 55 | 27 |

AUDIT-C, Alcohol Use Disorders Identification Test-Concise; PGSI, Problem Gambling Severity Index.

Observational Studies in Epidemiology guidelines for cross-sectional studies.

Characteristics of the sample and descriptive statistics are presented using weighted descriptive statistics. Under the first study aim, the prevalence of past-year gambling and any-risk gambling (according to the Problem Gambling Severity Index) are presented weighted with 95% CIs. Estimates are reported both as a percentage of the overall population, and of those who gambled in the past-year (in the case of any-risk gambling).

Under the second study aim, we constructed logistic regression models to assess the associations between any-risk gambling (reference group: no risk) with current smoking (reference group: not smoking), increasing

**Table 3** Association between current cigarette smoking, or drinking at increasing and higher risk levels, and any risk gambling according to the Problem Gambling Severity Index

| Variable | Event rate | OR | 95% CI | P value |
|---|---|---|---|---|
| Current cigarette smoking | | | | |
| PGSI category | | | | |
| No risk | 125/897 (14%) | — | — | |
| Any risk | 10/57 (18%) | 0.80 | 0.35 to 1.66 | 0.57 |
| Drinking at increasing and higher risk levels | | | | |
| PGSI category | | | | |
| No risk | 363/885 (41%) | — | — | |
| Any risk | 26/58 (45%) | 0.94 | 0.52 to 1.69 | 0.83 |

Model adjusted for age, sex, social grade and region.
PGSI, Problem Gambling Severity Index.

and higher risk alcohol consumption (reference group: <4 on Alcohol Use Disorders Identification Test-Concise (AUDIT-C)), respectively among people who gambled in the past year. All models are adjusted for key socio-demographic characteristics (age, sex, social grade and region). Respondents with missing data on any of the covariates of interest (3.4% of the total sample) were excluded from the analyses.

Under the third study aim, we estimated the average weekly expenditure on (1) gambling and (2) gambling, smoking and alcohol use, among those classified as 'any-risk' compared with those who gamble without risk. These data are presented descriptively as a mean expenditure with measures of spread (SE, 95% CI and range).

### Unregistered changes to the analysis plan
Observed non-significant associations between any risk of gambling harm and current smoking, or drinking at increasing and higher risk levels could have indicated evidence for no association, or that the data were insensitive to detect an effect. To explore this, post hoc Bayes factors are calculated for a range of hypothetical effect sizes including the potential for lower odds (OR=0.5 or 0.9) or higher odds (OR=1.1, 1.5 or 1.9) of any risk of gambling harm according to cigarette smoking and AUDIT-C scores of 4 or more, respectively. Bayes factors were computed using an online calculator (www.bayes-factor.info).

### RESULTS
A weighted total of 2398 adults aged 18 and older (mean (SE) age=47.7 (0.46)) surveyed in October 2022 were included in the analytical sample (table 2). In the overall sample, 43.6% (95% CI 41.2% to 45.9%) of adults participated in a gambling activity in the past year, and 3.2% (95% CI 2.3% to 4.1%) were classified as having any-risk of harm from gambling (ie, scoring >0 on the PGSI), 14.5% (95% CI 12.8% to 16.3%) were currently smoking and 33.4% (95% CI 31.2% to 35.7%) were drinking at increasing and higher

risk levels. Among those who reported any gambling activity in the past year (n=1045), 7.3% (95% CI 5.3% to 9.3%) were classified as being at any-risk of harm from gambling with 0.3% (95% CI 0.0% to 0.66%) classified as disordered gambling, 16.0% (95% CI 13.2% to 18.8%) were currently smoking and 40.8% (95% CI 37.2% to 44.4%) were drinking at increasing and higher risk levels (table 2). Aside from gambling on the national lottery, other lotteries or scratch cards (38.4%), the three most common gambling activities overall were online betting with bookmaker (5.5%), horse races (not online) (4.8%) and online gambling in slots, casino or bingo (4.1%) (online supplemental table S1).

In the models adjusting for age, sex, social grade and region, there were no apparent associations between any risk of harm from gambling and current cigarette smoking (OR adjusted=0.80, 95% CI 0.35 to 1.66) or drinking at increasing and higher risk levels (OR adjusted=0.94, 95% CI 0.52 to 1.69), respectively (table 3 and online supplemental table S2). Analyses using Bayes factors indicated that the data were insensitive to detect an effect in either direction, and therefore these results are inconclusive (online supplemental table S3).

In the sample of adults who gambled in the past year, the mean weekly spend on gambling was £4.80 (95% CI £4.18 to £5.43) among those classified as at no risk, and £45.68 (95% CI £12.07 to £79.29) among those classified as at any risk of harm from gambling according to the PGSI (figure 1 and online supplemental table S4). Caution should be taken in the interpretation of expenditure in the any-risk category due to a relatively small number of cases (n=67) compared with the no-risk category (n=878). The distribution of mean weekly spend on gambling is shown in online supplemental figure S1 and highlights how the mean is influenced by a small number of higher values in the any-risk category (one respondent reported a weekly mean spend on gambling of £998.00). The equivalent

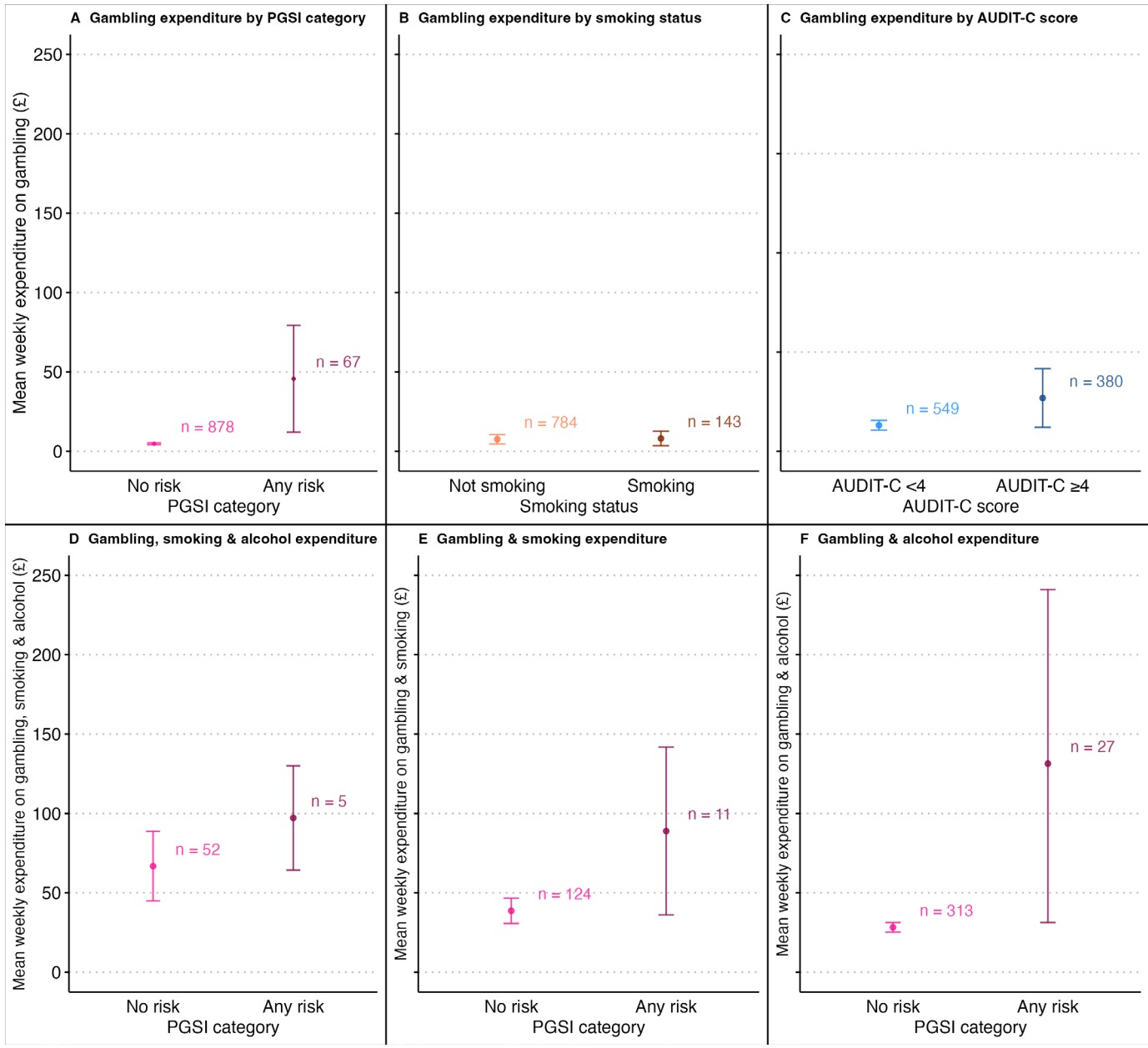

**Figure 1** Mean weekly expenditure on gambling according to PGSI category (A) smoking status (B) and AUDIT-C score (C) and mean weekly expenditure on gambling, smoking and alcohol (D) gambling and smoking (E) and gambling and alcohol (F) according to PGSI category. AUDIT-C, Alcohol Use Disorders Identification Test-Consumption; PGSI, Problem Gambling Severity Index.

expenditure in a sample excluding those who only gambled on lottery/scratch cards was £6.42 (95% CI £4.99 to £7.87) in those at no risk and £56.48 (95% CI £14.79 to £98.17) in those at any risk of harm from gambling. The mean weekly spend on gambling was £8.09 (95% CI £3.52 to £12.65) in people currently smoking (vs £7.61 in those not smoking) and £10.74 (95% CI £4.86 to £16.66) among people drinking at increasing and higher risk levels (vs £5.26 in people with AUDIT-C scores of <4), respectively (figure 1 and online supplemental table S4). Overall, among those who smoked or were drinking at increasing and higher risk levels, spending on gambling and smoking

was £42.73 (95% CI £33.88 to £51.59.), gambling and alcohol was £36.48 (95% CI £26.83 to £46.13), and on all three behaviours was £69.37 (95% CI £48.78 to £89.96) (online supplemental table S4).

Error bars represent 95% CIs for estimates of mean weekly expenditure.

## DISCUSSION
In a nationally representative survey of smoking and alcohol use in Great Britain, our pilot gambling questions collected during 1 month of data collection returned similar estimates for overall gambling participation, and

for at-risk and disordered gambling as other national population surveys.[1 8] Approximately half of adults reported some gambling activity in the past year, and descriptively the prevalence of smoking (16%) and increasing and higher risk drinking (41%) was higher in those who gambled compared with those who did not (13% and 28%, respectively).

One in 14 people who gambled were classified as being at any risk of harm from gambling, but our data were likely insensitive to detect associations between any-risk of harm from gambling, smoking and higher risk drinking, if true associations ranging from OR=0.5 to 1.9 existed. Although gambling at any level of harm is consistently associated with alcohol consumption,[2 29] the aetiology of this relationship is likely multidimensional. Observed associations in the wider literature may reflect common underlying genetic, social and environmental determinants,[30 31] but also involve bi-directional causality whereby the frequency of gambling is higher under the influence of alcohol.[30] Alcohol may be used as an avoidant coping mechanism following losses[32] and as a cued response following wins.[33] The mechanism through which alcohol consumption might lead to disordered gambling requires more research. For example, a recent review examining the salient hypothesis that acute alcohol consumption leads to harm from gambling by encouraging greater risk taking concluded that there was no reliable effect.[34]

In 2009 a review into the comorbidity of smoking and gambling concluded that comorbidity was highly prevalent.[13] However, an evidence review published in 2023 conducted by Public Health England concluded that cigarette smoking was not associated with gambling among adults.[2] While this may not hold true in certain priority subgroups, further data examining this issue in the STS/ATS could provide important information at the population-level.

Limitations of this study include the data being cross-sectional and self-reported, and the uncertainty in our estimate for the prevalance of any-risk of harm from gambling due to the relatively small number of respondents classified into this group. The paucity of data in our survey on individuals experiencing disordered gambling reflects the relatively small proportion of the population falling into this category, but also that population surveys cannot comprehensively capture relatively rare behaviours—like disordered gambling,[35] or injecting drug use[36]—which are more common in subgroups of the population who fall outside of traditional sampling frames.[37] Nonetheless, because they are more numerous, the majority of harms from gambling at the population-level is driven by those classified as low, and moderate risk of harm.[5] Understanding the relationships between gambling behaviour and other licit and commercially influenced addictive behaviours like smoking and alcohol use, and gambling at all levels of risk remains an important endeavour. While pilot data were collected in one survey month, extending data collection in a longer monthly time-series would allow these intersections to be interrogated with greater accuracy and reliability.

Finally, while there was wide uncertainty in our estimates for expenditure on gambling among those at any risk of harm from gambling, the outliers driving this uncertainty remain plausible given the extreme spending that can occur in those experiencing disordered gambling.[5] Indeed, due to the rise in online gambling in recent years, in their recent white paper the UK government has proposed introducing financial risk-checks for moderate to high spending.[38] While our estimates should be interpreted with caution, there was a signal for higher expenditure on gambling among those categorised as drinking at an increasing and higher risk level. If true, this pattern of spending would conform to studies highlighting a positive relationship between increasing alcohol consumption and gambling spend.[9]

In conclusion, the collection of pilot data on gambling in a population-level survey on smoking and alcohol use yielded estimates of gambling participation and at-risk and disordered gambling that are similar to other population-level surveys. Further data collection would help elucidate the intersections more reliably between gambling, smoking and alcohol use and inform population-level approaches to reduce the harms to public health conferred by these behaviours.

**Contributors** LK is guarantor for this work, had full access to all of the data in the study and took responsibility for the integrity of the data and the accuracy of the data analysis. LK was in charge of acquisition, analysis and interpretation of data, drafting of the manuscript, statistical analysis. All authors (LK, JB, SC, LS, VB, SS and AR) were involved in the concept and design of the study. LK led the revision of the manuscript, and all authors were involved in the revision of the manuscript for important intellectual content. LK and JB obtained funding, and JB was LK's supervisor.

**Funding** Funding for STS/ATS data collection is supported by Cancer Research UK (PRCRPG-Nov21\100002), and the UK Prevention Research Partnership (MR/S037519/1).

**Competing interests** JB reports receiving grants from Cancer Research UK during the conduct of the study and receiving unrestricted research funding from pharmaceutical companies who manufacture smoking cessation medications to study smoking cessation outside the submitted work. LS reports receiving honoraria for talks, receiving an unrestricted research grant and travel expenses to attend meetings and workshops by pharmaceutical companies that make smoking cessation products (Pfizer and Johnson & Johnson), and acting as a paid reviewer for grant-awarding bodies and as a paid consultant for health care companies. SC has provided expert consultancy to providers of UK life insurance and the pharmaceutical industry on matters relating to smoking cessation aids. SS has received funding from the Society for the Study of Addiction (SSA), and the NIHR. He is currently employed at the National Addictions Centre, part of the NIHR Biomedical Research Centre and declares no conflicts. AR has received funding from Santander, Public Health for Lincoln, The Royal Society, The Maurice and Jacqueline Bennett Charitable Trust, East Midlands RDS and internal University of Lincoln awards and declares no conflicts of interest. LK, VB and EP have no conflicts of interests to declare.

**Patient and public involvement** Patients and/or the public were not involved in the design, or conduct, or reporting, or dissemination plans of this research.

**Patient consent for publication** Consent obtained directly from patient(s).

**Ethics approval** Ethical approval for the STS/ATS was granted by the UCL Ethics Committee (ID 0498/001). The data were not collected by UCL and were anonymised when received by the research team. Participants gave informed consent to participate in the study before taking part.

**Provenance and peer review** Not commissioned; externally peer reviewed.

**Data availability statement** Data are available upon reasonable request. The analyses were preregistered on the open science framework, https://osf.io/nc6jm and conducted in R V.4.2.2 (packages tidyverse and survey) with all statistical code made open-access at https://osf.io/aj7c9/. Data are available from authors upon request.

**ORCID iD**
Loren Kock http://orcid.org/0000-0002-2961-8838

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
