## [Reviewer comments · BMJ Open]

ARTICLE DETAILS

TITLE (PROVISIONAL)	The Intersection of Gambling with Smoking and Alcohol use in Great Britain: A Cross-Sectional Survey in October 2022
AUTHORS	Kock, Loren; Cox, Sharon; Shahab, Lion; Roberts, Amanda; Sharman, Steve; Buss, Vera; Brown, Jamie

VERSION 1 – REVIEW

REVIEWER	Kesaite, Viktorija University of Glasgow
REVIEW RETURNED	13-Oct-2023

GENERAL COMMENTS	The study used a recent data (based on 2022) to assess gambling, smoking, and alcohol relationships. It's good to see that the authors followed STROBE checklist for cross-sectional studies. There are some specific suggestions for the study: - The motivation behind this study needs further development. Make it clearer in the abstract/introduction on why it's important to assess these relationships. A related study that you might want to draw on: Hing, N. and Russell, A.M., 2020. Proximal and distal risk factors for gambling problems specifically associated with electronic gaming machines. Journal of Gambling Studies, 36(1), pp.277-295.- The description of the outcomes (in the abstract and elsewhere) could be improved. What are the key outcomes? Are they continuous, binary etc? What is the key exposure (s)?- It's good to see recent data, however, is this data representative of the overall population i.e., is there an issue with sample selection bias? What are the main limitations? Are these variables self-reported etc?- Under the 'measures' (page 4), there is too much unnecessary detail on each measure. A table summary of key definitions of outcomes & exposures would improve the presentation.- Check some of the wording and ensure that it's referenced appropriately (e.g., lines 33-34 on page 3).- There should be a more detailed discussion on the appropriate methodology that is used to assess the research questions.
--

REVIEWER	Wilson, Charley Liverpool John Moores University
REVIEW RETURNED	03-Jan-2024

GENERAL COMMENTS	Thank you to the authors for providing this article, on an important topic for population health. The article is well written and it is especially promising to see that relevant gambling measures were included in a national population level survey.
--

	Introduction: In the opening paragraph of the introduction the overview of the distribution of gambling harms could have been strengthened by noting that the majority of overall harms are from those below the threshold for 'gambling disorder'. However, it is noted that this is highlighted in the discussion with reference 36. Another study that may be of interest to the authors to cite in the current article is: https://link.springer.com/article/10.1007/s10899-019-09902-8. This paper utilises household survey data from a region of GB and examines associations between low risk and moderate-risk/'problem gambling' (defined using the PGSI) and relevant health risk behaviours - including smoking and drinking alcohol. In paragraph three of the introduction the impacts of co-occurring gambling/smoking/alcohol use is discussed well in relation to those on lower incomes, however, this would be strengthened by briefly discussing the impacts that this may then have on health inequalities experienced between those on low/middle/higher incomes. Methods: For the measure of past year gambling - just for clarity was there an option in this question for respondents to say 'I haven't participated in gambling in the past year' or if they did not participate in gambling did respondents just not select any of the options? Results: In text when adjusted odds ratios are mentioned it would be useful to state in text which variables were controlled for. This is already done by the relevant table for these results, however, having it in text would help to have clarity while reading. The paragraph on spending was slightly confusing: 'According to smoking and drinking behavior, the mean weekly spend was £8.09 (3.52-12.65) in people currently smoking (vs. £7.61 in those not smoking) and £10.74 (4.86-16.66) among people drinking at increasing and higher risk levels (vs. £5.26 in people with AUDIT-C scores of < 4), respectively.' Was this talking about spending on gambling still? If so it would be clearer if this was explicitly stated, as there was some slight confusion whether this was relating to gambling spend or spend on smoking/drinking alcohol. In relation to the table looking at spending, it was not entirely clear why are there only a small number of respondents included in analyses on total alcohol/smoking/gambling spend? This measure would also be very interesting to see any future/further analyses on. Discussion: The discussion is well written and clear and puts the findings into the context of gambling measures being piloted successfully in a national population level survey, and strengthens the case for their future inclusion. Many thanks.
--	---

VERSION 1 – AUTHOR RESPONSE

1	The motivation behind this study needs further development. Make it clearer in the abstract/introduction on why it's important to assess these relationships. A related study that you might want to draw on: Hing, N. and Russell, A.M., 2020. Proximal and distal risk factors for gambling problems specifically associated with electronic gaming machines. Journal of Gambling Studies, 36(1), pp.277-295.	Thank you. We have updated the introduction to try and clarify the motivation for exploring co-occurrence of harmful gambling, smoking and higher risk alcohol consumption. The following paragraph has been updated and extended as follows: “The co-occurrence of gambling with smoking and increasing and higher risk alcohol consumption is important to study at the population level in the context of public health and health inequalities. First, it is likely that co-occurrence of these behaviours compound the physical, social, financial and psychological harms that each of them cause. These harms may be disproportionately greater for certain sub-groups, namely those experiencing poverty [16] who are also more likely to smoke and experience greater harms from alcohol consumption compared with more advantaged groups [17,18]. Second, due to their high relative costs, expenditure on smoking and alcohol can exacerbate and push low-income households into poverty [14]. Likewise, money spent on gambling as a proportion of total expenditure may be higher in less advantaged households [19]. Expenditure on all three products is of concern particularly in the UK and elsewhere where rising prices for everyday items and services have resulted in less ‘disposable’ income,
---	--	--

		particularly among lower income households [20]. Less able to absorb the added burden of this 'cost of living' crisis, it follows that less advantaged groups suffer greater psychosocial and material harm than more advantaged households even in the absence of the harms caused by gambling and substance use behaviours [21]. Smoking, alcohol and gambling among adults aged 18+ is currently legal in the UK, with the highly profitable underlying industries regulated to different degrees by the UK government but with similar motives to disrupt policies seeking to reduce the harm from use of their products [22]. Data from a representative sample of adults can provide insight into the dynamics of these behaviours – for instance the potential to substitute or complement one with another [23,24] – in an evolving sociocultural and regulatory context.
1	The description of the outcomes (in the abstract and elsewhere) could be improved. What are the key outcomes? Are they continuous, binary etc? What is the key exposure (s)?	The abstract has been revised slightly to make this clearer: We examined the prevalence of gambling in the past year and, among those reporting gambling, assessed the associations between the outcome of any risk of harm from gambling (scores of >0 on the problem gambling severity index) and the binary predictor variables of current cigarette smoking and higher risk alcohol consumption (AUDIT-C>4). We also explored the association of

		average weekly expenditure on gambling with smoking and alcohol use among those categorised at any-risk of harm from gambling. Also see below the updated measures section which we hope has addressed this concern.
1	It's good to see recent data, however, is this data representative of the overall population i.e., is there an issue with sample selection bias? What are the main limitations? Are these variables self-reported etc?	Yes, the data are representative of adults in Great Britain, and results are unlikely to suffer from selection bias. A summary of the survey methodology and indications of representativeness is provided in the sample and recruitment section: “The STS/ATS uses a hybrid of random location and quota sampling to select a new sample of approximately 2,400 adults (aged ≥16 years) each month in Great Britain. Telephone interviews are carried out with one household member until quotas based on factors influencing the probability of being at home (e.g., gender, age, working status) are fulfilled. We used survey weighting to match descriptive data to sociodemographic profile in Great Britain (based on age, social grade, region, tenure, ethnicity and working status within sex). Detailed survey methodology is reported elsewhere [17,18]. Comparisons with sales data and other national surveys show that the STS recruits a representative sample of the population in Great Britain with

		regard to key demographic variables and smoking indicators.” The main limitations of the survey (outlined in the discussion section) in the context of this study are that data are cross-sectional and self-reported (so we are unable to draw causal conclusions about the influence of one behavior on another) and that because a small percentage (0.1-0.5%) of the population in Great Britain are classified as at any-risk of harm from gambling, this results in few respondents (76) in our survey of approximately 2000 individuals.
1	Under the ‘measures’ (page 4), there is too much unnecessary detail on each measure. A table summary of key definitions of outcomes & exposures would improve the presentation.	Thank you, we agree that this could be improved, and have created new Table 1 which summarizes key information about the measures, including information on which are outcomes, predictors and covariates. The table also refers readers to the supplementary appendix for full item descriptions used in variable coding.
1	Check some of the wording and ensure that it’s referenced appropriately (e.g., lines 33-34 on page 3).	Thank you. We have rephrased and re-specified this section as follows: “Due to their high relative costs, expenditure on smoking and alcohol can exacerbate and push low-income households into poverty [14]. Likewise, money spent on gambling as a proportion of total expenditure may be higher in less advantaged households [19]. Expenditure on all three products is of concern

		particularly in the UK and elsewhere where rising prices for everyday items and services have resulted in less 'disposable' income, particularly among lower income households [20].”
1	There should be a more detailed discussion on the appropriate methodology that is used to assess the research questions.	We have updated the analysis section to more clearly indicate which study aim each analysis is addressing.
2	In the opening paragraph of the introduction the overview of the distribution of gambling harms could have been strengthened by noting that the majority of overall harms are from those below the threshold for 'gambling disorder'. However, it is noted that this is highlighted in the discussion with reference 36.	We agree that this is an important point that could be made earlier. We have updated text in the first paragraph: “Conservative estimates indicate that approximately 0.3-0.5% of the general UK adult population report severe gambling behaviours that warrant a diagnosis of gambling disorder (hereafter termed “disordered gambling”) and 3-4% are “at-risk” (those who experience a low or moderate level of problems leading to some negative consequences, and relative to disordered gamblers drive most of the harm from gambling at the population level) [2,5,6]”
2	Another study that may be of interest to the authors to cite in the current article is: Butler, N., Quigg, Z., Bates, R., Sayle, M., & Ewart, H. (2020). Gambling with your health: Associations between gambling problem severity and health risk behaviours, health and wellbeing. Journal of Gambling Studies , 36, 527-538.	Thank you for bringing this relevant study to our attention – we have cited it in the opening of the second paragraph.

	This paper utilises household survey data from a region of GB and examines associations between low risk and moderate-risk/'problem gambling' (defined using the PGSI) and relevant health risk behaviours - including smoking and drinking alcohol.	
2	In paragraph three of the introduction the impacts of co-occurring gambling/smoking/alcohol use is discussed well in relation to those on lower incomes, however, this would be strengthened by briefly discussing the impacts that this may then have on health inequalities experienced between those on low/middle/higher incomes.	In line with this comment and that of reviewer 1, we have updated this paragraph: “Due to their high relative costs, expenditure on smoking and alcohol can exacerbate and push low-income households into poverty [15]. Likewise, money spent on gambling as a proportion of total expenditure may be higher in less advantaged households [20]. Expenditure on all three products is of concern particularly in the UK and elsewhere where rising prices for everyday items and services have resulted in less ‘disposable’ income, particularly among lower income households [21]. Since these individuals are less able to absorb the added burden of this ‘cost of living’ crisis, it follows that less advantaged groups suffer greater psychosocial and material harm than more advantaged households even in the absence of the harms caused by gambling and substance use behaviours [22]. Smoking, alcohol and gambling among adults aged 18+ is

		currently legal in the UK, with the highly profitable underlying industries regulated to different degrees by the UK government but with similar motives to disrupt policies seeking to reduce the harm from use of their products [23]. Data from a representative sample of adults can provide insight into the dynamics of these behaviours – for instance the potential to substitute or complement one with another [24,25] – in an evolving sociocultural and regulatory context.”
2	Methods: For the measure of past year gambling - just for clarity was there an option in this question for respondents to say 'I haven't participated in gambling in the past year' or if they did not participate in gambling did respondents just not select any of the options?	Thank you. Yes, there was an option “ Have not done any of these”. The new measures table has been updated with this information.
2	Results: In text when adjusted odds ratios are mentioned it would be useful to state in text which variables were controlled for. This is already done by the relevant table for these results, however, having it in text would help to have clarity while reading.	This has been added to the text.
2	The paragraph on spending was slightly confusing: 'According to smoking and drinking behavior, the mean weekly spend was £8.09 (3.52-12.65) in people currently smoking (vs. £7.61 in those not smoking) and £10.74 (4.86-16.66) among people drinking at increasing and higher risk levels (vs. £5.26 in people with	Thank you for highlighting this. We have added some clarifying text at the start of the sentence as follows: “The mean weekly spend on gambling was £8.09 (3.52-12.65) in people currently smoking (vs. £7.61 in those not

	AUDIT-C scores of <4), respectively.' Was this talking about spending on gambling still? If so it would be clearer if this was explicitly stated, as there was some slight confusion whether this was relating to gambling spend or spend on smoking/drinking alcohol.	smoking) and £10.74 (4.86-16.66) among people drinking at increasing and higher risk levels (vs. £5.26 in people with AUDIT-C scores of < 4), respectively."
2	In relation to the table looking at spending, it was not entirely clear why are there only a small number of respondents included in analyses on total alcohol/smoking/gambling spend? This measure would also be very interesting to see any future/further analyses on.	The question on expenditure on cigarettes is asked only to those who are currently smoking, and the question on alcohol expenditure is asked only of those who score >4 on the AUDIT items 1, 2, and 3. The calculation of expenditure on all three behaviours necessarily restricts the sample to the small number of respondents who provide these data on all three behaviours. To provide some more valuable information about expenditure in the context of smoking and drinking, we have now also calculated expenditure on gambling and smoking, and gambling and alcohol, in addition to expenditure on all three. To incorporate these data and summarize the expenditure results more clearly, a new 6 panel Figure 1 has been produced, which presents (using the same y axis scale to aid comparison) mean expenditure with 95% CIs for: A: Gambling expenditure by PGSI category B: Gambling expenditure by smoking status

		C: Gambling expenditure by AUDIT-C score D: Gambling smoking, and alcohol expenditure. E: Gambling and smoking expenditure F: Gambling and alcohol expenditure The previous table with informative data on the distribution of expenditure is now included as a supplementary table S4 and accompanying Figure S1. We believe this provides some insightful information, and we have added some text to the results to reflect this. The updated results paragraph on expenditure now reads: “In the sample of adults who gambled in the past year, the mean weekly spend on gambling was £4.80 (95% CI 4.18-5.43) among those classified as at no risk, and £45.68 (12.07-79.29) among those classified as at any risk of harm from gambling according to the PGSI (Figure 1 and Table S1). Caution should be taken in the interpretation of expenditure in the any risk category due to a relatively small number of cases (n=67) compared with the no risk category (n=878). The distribution of mean weekly spend on gambling is shown in Figure S1 and highlights how the mean is influenced by a small number of higher values
--	--	---

		in the any risk category (one respondent reported a weekly mean spend on gambling of £998.00). The equivalent expenditure in a sample excluding those who only gambled on lottery/scratch cards was £6.42 (4.99-7.87) in those at no risk and £56.48 (14.79-98.17) in those at any risk of harm from gambling. The mean weekly spend on gambling was £8.09 (3.52-12.65) in people currently smoking (vs. £7.61 in those not smoking) and £10.74 (4.86-16.66) among people drinking at increasing and higher risk levels (vs. £5.26 in people with AUDIT-C scores of < 4), respectively (Figure 1 and Table S4). Overall, among those who smoked or were drinking at increasing and higher risk levels, spend on gambling and smoking was £42.73 (33.88-51.59.), gambling and alcohol was £36.48 (26.83-46.13), and on all three behaviours was £69.37 (48.78-89.96) (Table S1).”
2	The discussion is well written and clear and puts the findings into the context of gambling measures being piloted successfully in a national population level survey, and strengthens the case for their future inclusion.	Thank you for your thoughtful review.

VERSION 2 – REVIEW

REVIEWER	Wilson, Charley Liverpool John Moores University
REVIEW RETURNED	04-Mar-2024

GENERAL COMMENTS

Thank you to the authors for making the relevant changes in line with reviewer comments. I found particularly the spending data interesting and far easier to digest with the amendments that have been made. Well done on the article.